# A Data-Centric Multi-Objective Learning Framework for Responsible Recommendation Systems

## ABSTRACT

Recommendation systems effectively guide users in locating their desired information within extensive content repositories. Generally, a recommendation model is optimized to enhance accuracy metrics from a user utility standpoint, such as click-through rate or matching relevance. However, a responsible industrial recommendation system must address not only user utility (responsibility to users) but also other objectives, including increasing platform revenue (responsibility to platforms), ensuring fairness (responsibility to content creators), and maintaining unbiasedness (responsibility to long-term healthy development). Multi-objective learning is a potent approach for achieving responsible recommendation systems. Nevertheless, current methods encounter two challenges: difficulty in scaling to heterogeneous objectives within a unified framework, and inadequate controllability over objective priority during optimization, leading to uncontrollable solutions.

In this paper, we present a data-centric optimization framework, MoRec, which unifies the learning of diverse objectives. MoRec is a tri-level framework: the outer level manages the balance between different objectives, utilizing a proportional-integral-derivative (PID)-based controller to ensure a preset regularization on the primary objective. The middle level transforms objective-aware optimization into data sampling weights using sign gradients. The inner level employs a standard optimizer to update model parameters with the sampled data. Consequently, MoRec can flexibly support various objectives while maintaining the original model intact. Comprehensive experiments on two public datasets and one industrial dataset showcase the effectiveness, controllability, flexibility, and Pareto efficiency of MoRec, making it highly suitable for real-world implementation.

## CCS CONCEPTS

• **Information systems** → **Recommender systems**.

## KEYWORDS

Recommender System, Multi-objective Learning, Data Sampling

**ACM Reference Format:**
Anonymous Author(s). 2024. A Data-Centric Multi-Objective Learning Framework for Responsible Recommendation Systems. In *Proceedings of The Web Conference 2024 (WWW '24)*. ACM, New York, NY, USA, 9 pages. https://doi.org/XXXXXXX.XXXXXXX

## 1 INTRODUCTION

Recommender systems play a crucial role in enhancing user experience and optimizing service providers' profits by selecting relevant items from a vast pool and presenting them to users. Over the past decade, there has been a growing research focus on technical advancements in recommender systems, particularly on deep learning techniques [14, 20, 21, 37, 39, 42, 46, 49]. Typically, recommender models are optimized for overall user utilities (referred to as overall accuracy hereinafter), such as click-through rate (CTR) or recall metrics based on historical behavior logs. However, real-world industrial recommender systems should fulfill additional responsibilities beyond overall accuracy, including balancing utilization for different groups (fairness responsibility), benefiting multiple stakeholders (revenue responsibility), and reducing popularity bias (long-term engagement responsibility). Due to conflicts between these responsibilities, recommender systems that solely optimize global accuracy may fall into an unhealthy state, with other objectives remaining far from satisfactory [13, 41, 43].

Therefore, leveraging multi-objective learning methods to achieve a desirable trade-off among multifaceted responsibilities is essential for industrial recommender systems. Multi-task learning, a form of multi-objective learning where each objective is framed as a learning task and a shared base model is trained to optimize multiple tasks simultaneously, has garnered widespread attention in both academia [23, 26, 35] and industry [22, 24, 31, 35]. Specifically, the controller balancing different tasks can be either predefined static weights [24, 45] or dynamic weights with a Pareto solver [22, 31]. Nonetheless, existing approaches struggle to incorporate a comprehensive set of objectives, with the most frequently addressed objectives in recommender literature being accuracy and revenue. Moreover, although these approaches can lead to Pareto-efficient solutions, the properties of such solutions remain uncontrollable, potentially resulting in a significant decline in accuracy to accommodate a revenue increase. In this paper, we seek for a more efficient and flexible approach that optimizes multiple objectives in a unified, end-to-end, and model-agnostic manner while allowing for controllability based on predefined priorities for various objectives.

To achieve our goal, we first consolidate various objectives crucial for industrial recommender systems into four fundamental forms: accuracy, revenue, fairness, and alignment, with detailed definitions provided in Section 3.2. We believe that the most commonly used objectives in recommender systems can be categorized into one of these fundamental forms. Given the difficulty of converting some objectives (such as fairness and alignment) into differentiable functions on individual data samples, we adopt an adaptive data re-weighting framework during the training process, which can provide a unified way to optimize all the aforementioned objectives. Our framework is inspired by *FairBatch* [33], a data sampling method designed to improve a model's fairness, such as equalized

odds. The primary advantage of this method is its ease of implementation, as it does not require any modifications to data preprocessing or model architecture. Although FairBatch was originally designed for fairness, we find that the core theory behind its implementation, based on signed gradient descent, can be extended to other objectives, resulting in a unified framework for optimizing multiple objectives simultaneously.

Owing to the inherent conflicts among different objectives, it is generally challenging for a single solution to achieve optimized status for all objectives simultaneously. A Pareto-efficient solution [23] is one where no other solution can outperform it across all objectives at once. However, Pareto-efficient solutions are not unique, and if not trained properly, the model may produce an undesirable outcome, such as excessively optimizing fairness while significantly compromising accuracy. Guiding the model training to generate desirable solutions is crucial but cannot be guaranteed or controlled with existing multi-task-based frameworks, such as static weighted sums of objectives [8] or multiple-gradient descent algorithms (MGDA)[10, 23, 35]. Therefore, we incorporate the concept of the Proportional-Integral (PI) controller [1], which is derived from automatic control theory [2], into our training framework. The PI controller uses the training status as feedback and can automatically adjust the weights of objective functions to prevent the resulting solution from deviating from a desirable outcome. The PI controller, together with a data sampler and a base model optimizer, constitutes our novel tri-level optimization framework MoRec: on the first level, an objective coordinator dynamically adjusts the priorities of different objectives; on the second level, a data sampler collects a batch of training instances based on data weights that reflects the optimization of each objective; on the third level, a traditional model optimizer updates model parameters with the training instances.

We conduct experiments on three real-world datasets, comprising two public datasets and one industrial dataset. The results demonstrate that MoRec is effective in harmoniously optimizing various objectives, capable of generating Pareto-efficient solutions over baseline methods, and controllable in terms of accuracy settings. Our major contributions can be summarized as follows:

- We propose MoRec, a data-centric framework designed to optimize multiple objectives simultaneously in a unified manner. MoRec can be seamlessly integrated into existing recommender systems training pipelines without altering the original backbone model and optimizer.
- We consolidate various objectives into four fundamental types and design tri-level organized components in MoRec to ensure that the optimization process is controllable, Pareto-efficient, and extensible to various objectives.
- We conduct experiments on three real-world datasets to demonstrate the effectiveness, Pareto-efficiency, and controllability of MoRec. Source code will be released upon the paper's acceptance.

## 2 PRELIMINARY

Let $\mathcal{U}$, $\mathcal{E}$, and $D$ represent the sets of users, items, and user-item interactions, respectively. Each interaction in $D$ can be denoted as $x_i = (u_i, e_i)$, signifying that user $u_i$ has interacted with item $e_i$. Generally, a recommendation model $f(\cdot|\theta)$ with parameters $\theta$ is trained to minimize the overall error of fitting on $D$. For example, binary cross-entropy loss [20] can be used as follows:

$$l_{acc}(x_i) = -y_i log f(x_i|\theta) - (1 - y_i)log(1 - f(x_i|\theta)) \quad (1)$$

Here, $y_i \in \{0, 1\}$ denotes the label of data sample $x_i$. When $y_i = 0$, it implies that $e_i$ is a negative sample for $u_i$. This represents an accuracy-oriented optimization. However, a responsible recommender system should consider multiple objectives. Recently, Fair-Batch [33] introduced a framework that addresses dual objectives from a data-centric perspective, employing a bilevel optimization approach. The fundamental process in FairBatch involves optimizing objectives based on dynamic weights ($\boldsymbol{w}$) assigned to data samples. Its most appealing benefit is that **it eliminates the need for modifications to the model and loss function. The only component that requires alteration is the dataloader**, which greatly enhances usability and convenience when upgrading existing single-objective systems to multi-objective systems.

Consider the optimization of equalized odds and accuracy as objectives in FairBatch for illustration. The goal of the equalized odds measure is to ensure that the prediction is independent of the sensitive attribute, conditional on the true label, thus reducing disparities between advantaged and disadvantaged groups. Let $g_i \in \mathcal{Z}$ represent the sensitive attribute of sample $x_i$. All samples are classified into $|\mathcal{Z}|$ groups based on their sensitive attributes. Denote the sampling weight of group $z$ as $w_z$. The bilevel optimization problem can be formulated as follows:

$$\min_{\boldsymbol{w}} \max_{z_i \neq z_j} \{|L^{z_i}(\theta_{\boldsymbol{w}}) - L^{z_j}(\theta_{\boldsymbol{w}})|\}, \text{ s.t. } \forall w_z > 0,$$
$$\theta_{\boldsymbol{w}} = \arg\min_{\theta} \sum_{z \in \mathcal{Z}} w_z L^z(\theta). \quad (2)$$

where $L^z(\theta)$ represents the average loss of samples in group $z$ and $\boldsymbol{w} = \{w_z\}$ denotes the group-wise sampling weights. The inner-level optimization of $\theta_{\boldsymbol{w}}$ corresponds to the conventional SGD-based model training procedure. To address the outer-level optimiztion on $\boldsymbol{w}$, FairBatch uses a signed gradient-based algorithm. Assume that $(i^*, j^*) = \arg\max_{(i,j)} |L^{z_i}(\theta_{\boldsymbol{w}}) - L^{z_j}(\theta_{\boldsymbol{w}})|$, the update rule of $\boldsymbol{w}$ is:

$$w_{i^*}^{(t+1)} = w_{i^*}^{(t)} - \alpha \cdot \text{sign}(L^{z_{j^*}}(\theta_{\boldsymbol{w}}) - L^{z_{i^*}}(\theta_{\boldsymbol{w}})),$$
$$w_{j^*}^{(t+1)} = w_{j^*}^{(t)} + \alpha \cdot \text{sign}(L^{z_{j^*}}(\theta_{\boldsymbol{w}}) - L^{z_{i^*}}(\theta_{\boldsymbol{w}})). \quad (3)$$

The justification for the rationality of Eq (3) can be found in Lemma 1 of FairBatch, with the notable difference being that $w_j$ in this context is the sum of multiple $\lambda_i$ values in FairBatch. Intuitively, the update rule raises the sampling probability for a disadvantaged group while reducing it for an advantaged group. Inspired by Fair-Batch, we design a data-centric multi-objective learning framework for optimizing diverse objectives simultaneously in recommender systems.

## 3 METHODOLOGIES

### 3.1 Limitations of FairBatch

However, there are two major limitations when applying FairBatch to multi-objective recommender systems. Firstly, FairBatch only considers two objectives, namely fairness and accuracy. Recommender systems require more realistic objectives to be taken into

account, such as revenue, fairness, and unbiasedness. Secondly, Fair-Batch places excessive emphasis on the optimization of the fairness objective. It lacks a comprehensive discussion on balancing multiple objectives and controlling real-world constraints. For example, accuracy is a critical commercial metric, and it is often necessary to maintain similar performance levels when transitioning from single-objective systems to multi-objective-aware systems.

To overcome these limitations, we introduce MoRec, a trilevel data-centric framework designed to unify diverse objectives while offering controllablity. This framework comprises three interconnected levels that collaborate harmoniously, ensuring an efficient and effective process:

- **Outer-level - Objective Coordinator**: It coordinates the relationship between goals by monitoring and adjusting the objectives to achieve a desired balance and performance.
- **Middle-level - Adaptive Data Sampler**: Based on the coordinating signals from the outer-level, this component is in charge of updating sample weights and dynamically selecting training samples.
- **Inner-level - Standard Model Optimizer**: This is a standard optimizer such as SGD, concentrating on the training of the backbone model with selected data samples.

### 3.2 Foundation Objectives

To effectively capture a broad spectrum of objectives, we emphasize four core objectives to maximize: accuracy, revenue, fairness, and alignment. These categories are designed to encompass the majority of significant objectives for recommender systems. This approach enables the unification of various objectives optimization within a single data sampling framework, which can be implemented as a flexible plug-in data loader. In the following subsection, we will discuss each objective along with its respective update rules for the sampling weights $\boldsymbol{w}$. These rules are crucial for the data sampler's optimal performance and overall effectiveness.

***Accuracy***. This is the fundamental objective, formulated as the negative accuracy loss as shown in Eq.(4). Intuitively, we set the sampling weights as a uniform distribution, i.e., $w_i^{acc} = \frac{1}{|D|}$, which is consistent with the standard accuracy-oriented model's data loading process.

$$O_{acc} = -L(\theta) = -\frac{1}{|D|} \sum_{i=1}^{|D|} l_{acc}(x_i, \theta) \qquad (4)$$

***Revenue***. Industrial recommendation systems typically need to consider the revenue generated for the platform. In this case, each item $e_i$ is associated with a profit value $r(e_i)$, and the objective is to maximize the expected revenue of the recommended items:

$$O_{rev} = \frac{1}{|D|} \sum_{i=1}^{|D|} r(e_i) \cdot p(e_i|u_i)$$

Where $p(e_i|u_i)$ represents the likelihood of user $u_i$ accepting item $e_i$, which can be approximated by the negative loss $-l_{acc}(\cdot)$[1]. Consequently, the objective can be reformulated as maximizing the

---

[1]Explanation: In the case of the revenue objective, we regard each data sample as a positive item to be recommended. Thus, only the first part in Eq.(1) remains, and $l_{acc}(x_i) = -\log f(x_i|\theta)$.

following goal:

$$O_{rev} = -\frac{1}{|D|} \sum_{i=1}^{|D|} r(e_i) \cdot l_{acc}(x_i) \qquad (5)$$

Hence, for this objective, we set the weight of data sample $x_i = (u_i, e_i)$ as $w_i^{rev} \propto r(e_i)$ and maintain fixed. Note that the data weights for both the accuracy metric and revenue metric are constants that do not require update rules.

***Fairness***. Fairness pertains to the performance disparity across groups, such as whether there is unfairness in the recommendation accuracy for various genders or item categories. There are multiple definitions of fairness measurement in recommendation systems, and in this paper, we adopt the Least Misery [43] as the measurement. The least misery is denoted as the accuracy in the worst-performing group. The objective can be represented as maximizing the following goal:

$$O_{fai} = \max\left\{O_{acc}^z, \forall z \in \mathcal{Z}\right\} = \min\left\{L^z, \forall z \in \mathcal{Z}\right\}, \qquad (6)$$

where $O_{acc}^z$ and $L^z$ represent the average accuracy measure and loss of group $z$, respectively. To maximize the objective, we formalize the problem as following bilevel optimization:

$$
\begin{aligned}
\boldsymbol{w}^{fai} &= \arg\min_{\boldsymbol{w}} \max_{z \in \mathcal{Z}}\{L^z(\theta_{\boldsymbol{w}})\}, \ \text{s.t.} \ \forall w_z > 0, \\
\theta_{\boldsymbol{w}} &= \arg\min_{\theta} \sum_{z \in \mathcal{Z}} w_z L^z(\theta).
\end{aligned}
\qquad (7)
$$

The update rule of $\boldsymbol{w}^{fai}$ is as follows:

$$w_{i^*}^{(t+1)} = w_{i^*}^{(t)} - \alpha \cdot \text{sign}(0 - L^{z_{i^*}}(\theta_{\boldsymbol{w}^{fai}})) = w_{i^*}^{(t)} - \alpha \cdot (-1). \quad (8)$$

where $i^* = \arg\max_i L^{z_i}(\theta_{\boldsymbol{w}})$. Intuitively, the update rule elevates the sampling probability of the most disadvantaged group, i.e. group with the largest accuracy loss. Notably, $w_i^{fai}$ is initialized as proportional to number of samples in group $i$.

***Alignment***. Machine learning-based models tend to involve skew distributions in their predictive patterns. A typical phenomenon is bias amplification, in which some patterns are over-amplified in the learned model. For example, popularity amplification means that the model recommends too many popular items to users so that long-tail items have no chance to get exposed. To address the skew distribution issue, we propose the alignment objective, which aligns the model's distribution with some pre-defined expectation distribution:

$$O_{ali} = D_{KL}(Q||P) = \sum_{i=1}^{|D|} Q(x_i) \log \frac{Q(x_i)}{P(x_i, \theta)} \qquad (9)$$

Where $P(\cdot, \theta)$ denotes the output distribution of the model and $Q(\cdot)$ represents the predefined distribution to be followed. Without loss of generality, we use item popularity for illustration and experimentation. For sample $x_i = (u_i, e_i)$, we set the exposure volume of item $e_i$ in the model's recommendation list as $P(x_i, \theta)$ and the frequency of $e_i$ in the training data as $Q(x_i)$ (note that $Q(x_i)$ can be freely designated according to real demands).

 

**Figure 1: MoRec Framework.**

(a) Data Sampler      (b) Objective Coordinator

**Figure 2: Primary components in MoRec.**

As for the update of $w^{ali}$ to minimize $O_{ali}$, the bilevel optimization is formalized as follow:

$$w^{ali} = \arg\min_{w} \sum_i Q(x_i) \log \frac{Q(x_i)}{P(x_i, \theta_w)}, \text{ s.t. } \forall w_z > 0,$$
$$\theta_w = \arg\min_{\theta} \sum_{i=1}^{|D|} w_i l_{acc}(x_i, \theta). \tag{10}$$

The update rule of $w^{ali}$ is as follows:

$$w_i^{(t+1)} = w_i^t - \alpha \cdot \text{sign}\big(P(x_i, \theta_w) - Q(x_i)\big), \forall i \in \mathcal{Z} \tag{11}$$

which aims to adjust the weights of samples accordingly when they deviate from a desired level. And the initial value of $w_i^{ali}$ is set to $1/|D|$.

### 3.3 Objectives Coordinator

Coordinating multiple objectives presents a key challenge. [22] generates a Pareto Frontier by setting different bounding values for the objectives, and then selects the most suitable solution from the Pareto Frontier according to real business demands. [23, 26] utilize preference vectors to generate a well-distributed set of Pareto solutions to choose from, representing different trade-offs among objectives. In our initial implementations, we prioritized both bounding values and preference vectors. However, we soon realized that they failed to demonstrate any advantage over the naive linear scalarization method (Experimental results can be found in Section 4.4). A possible reason is that the unified optimization through data sampling homogenizes the gradients of training supervision, facilitating the effectiveness of linear scalarization. Consequently, we ultimately selected linear scalarization for its simplicity and effectiveness.

Specifically, let the vector of weights associated with each objective be denoted as $\rho = [\rho_{obj_1}, \ldots, \rho_{obj_n}]$, and let the losses of

objectives be represented by $\ell = [\ell_{obj_1}, \ldots, \ell_{obj_n}]$, with $n$ indicating the total number of objectives. The combined loss can then be calculated as $\ell_{final} = \rho^T \ell$. We can generate distributed solutions by varying the values of $\rho$.

On the other hand, it is often important to ensure that crucial objectives, such as accuracy, do not significantly deteriorate when optimizing multiple objectives, thereby enabling a safe and smooth system upgrade. To achieve this, we draw inspiration from a method in the field of control systems – the PID (Proportional-Integral-Derivative) controller[1]. The PID is a widely-used feedback loop component in industrial control applications, designed to regulate a specific performance metric of the system to a predetermined value. This property aligns well with our objective of maintaining the model's accuracy at a desirable level, and is thus adopted for our objectives coordinator. In the new setting, the coefficient of the accuracy objective $\rho_{acc}$ is no longer a preset, fixed value, but rather an adaptively changing one. To emphasize this, we rewrite $\ell_{final}$ as follows:

$$\ell_{final} = \alpha_{acc} \cdot \ell_{acc} + \sum_{obj \neq acc} \alpha_{obj} \cdot \ell_{obj}. \tag{12}$$

The key aspect here is determining how to adjust $\alpha_{acc}$. Inspired by ControlVAE [36], we remove the derivative term in PID, and regard $\ell_{acc}$ as the performance metric to be controlled. Loss value $\hat{\ell}$ represents the final stable value of loss in a model optimized for a single objective, it servers as the the preset value to control $\ell_{acc}$ in PI. Denote the loss value of the model at time $t$ as $l^{(t)}$, then the output of Objective Coordinator $OC\big(\ell^{(t)}; \rho, \hat{\ell}\big)$ is:

$$\alpha_{obj}(t) = \begin{cases} \rho_{obj}, & obj \neq acc; \ (13) \\ \dfrac{K_p}{1 + \exp(err(t))} - K_i \sum_{j=1}^{t} err(j) + \alpha_{min}, & obj = acc. \ (14) \end{cases}$$

where $err(t) = \hat{\ell} - \ell^{(t)}$ denotes the error at time $t$, i.e, the model's accuracy performance gap on the current mini-batch of samples. $K_p, K_i$ are the non-negative coefficients of the proportional and integral parts, respectively, and $\alpha_{min}$ is a constant reflecting the minimum value. These three variables are hyper-parameters.

The core idea of PI equation (Eq.(14)) is to apply a correction in the direction to reduce the error between the preset loss value and the current loss value. Specifically, the first term (proportional term, abbreviated as the P-term) controls the accuracy metric in the current mini-batch of samples. When $err(t)$ is negative and its absolute value is large, it indicates that the data samples are currently poorly fitted. Consequently, the P-term would be increased, promoting the model to strengthen the learning on $\ell_{acc}$. Conversely, a larger positive $err(t)$ indicates that the model may overfit those samples, prompting the P-term to decrease and reducing the weight of accuracy loss. This adjustment allows more room for the model to optimize other objectives. The second term (integral term, abbreviated as the I-term) manages accuracy from the perspective of cumulative errors, which essentially reflect the overall trend across the entire dataset rather than focusing solely on the current mini-batch's samples. Note that $\sum_t err(t)$ represents the average error across all samples. If the value is positive, it suggests that the average loss is smaller than the preset value, which may potentially lead the model into an unexpected state, such as overfitting.

In this case, the I-term would reduce the weight on the control metric $\ell_{\text{acc}}$. Conversely, if the model has not reached the preset state in terms of average loss, i.e., $\sum_t err(t) < 0$, the I-term will be positive, assigning a stronger weight to the control metric $\ell_{\text{acc}}$. This approach stabilizes the model's accuracy performance to the preset value throughout the training process, resulting in enhanced controllability.

### 3.4 Overall Framework

The overall framework of MoRec is illustrated in Figure 1, while Figure 2a and Figure 2b present an enlarged view of the adaptive data sampler and the objective coordinator. Meanwhile, we offer an algorithmic pseudo-code in Algorithm 1. In summary, first, the objective coordinator is initialized with the preset objective priority $\rho$ and $\hat{\ell}$, responsible for loss synthesis in line 6, denoted as the outer level. Then the sampling weights $w$ are initialized with the training and validation set, which is mentioned in Section 3.2. Mini-batches are dynamically sampled according to weights $w$ in line 3, and the weights $w$ are optimized on the validation set $D_v$ after each training epoch in line 9, comprising the middle level. Finally, loss values corresponding to various objectives are calculated in line 5 and are synthesized with the output of the objective coordinator. The model's parameters $\theta$ are updated with the synthesized loss in line 8 as the inner level.

---

**Algorithm 1:** The Tri-level Framework MoRec.

---

**Input:** Training Data $D_t$, Valid Data $D_v$, Data Sampler DS, Objective Coordinator OC, Objective priority vector $\rho$, Expected accuracy loss $\hat{\ell}$

**Output:** Model $\theta$.

1 Initialize sampling weight $w$ in DS with $D_t$ and $D_v$;

2 **repeat**

3      Draw minibatches from $D_t$ according to sampling weight $w$;    // Middel Level: data sampling

4      **for** batch $\in$ minibatches **do**

5          $\ell \leftarrow$ Calculate loss with $\theta$;

6          $\alpha \leftarrow$ OC$(\ell; \rho, \hat{\ell})$ according to Eq. (14);    // Outer Level: objective control

7          loss $\leftarrow \alpha^T \ell$;

8          Update $\theta$ with loss;    // Inner Level: model optimization

9      Update sampling weight $w$ in DS with $D_v$ according to Eq. (8) and Eq. (11);    // Middel Level: sampling weight update

10 **until** *Convergence or reaching max epoch*;

---

## 4 EXPERIMENT

### 4.1 Experimental Setting

*4.1.1 Dataset.* We evaluate our method on three real-world datasets, including two public Amazon[2] datasets and one industrial dataset provided by Xbox. The basic statistics are illustrated in Table 1. The Electronics and Movies datasets contain user reviews of products on the Amazon platform, with ratings ranging from 0 to 5. We filter out reviews with ratings below 3. The Xbox dataset consists of records of users' purchase behaviors on video games. For all datasets, we apply the K-core filtering technique, setting K to 5, to obtain high-quality data. As for the dataset partitioning, we utilize the widely adopted leave-one-out method, which is prevalent in evaluating recommender models. We reserve the most recent interaction of each user for the test set and use their second most recent

---

[2]http://jmcauley.ucsd.edu/data/amazon/links.html

interaction for validation purposes, while allocating the remaining items for training.

We examine four distinct objectives - accuracy, revenue, popularity alignment, and fairness. However, constructing multiple objectives necessitates the use of side information beyond mere interaction data. As a result, we leverage item category and price attributes to facilitate this process. To compute the fairness metric, we utilize item category information to divide items into distinct groups. In addition, we employ item prices as an indicator to estimate the platform's profit from recommendations for revenue estimation. Lastly, to avoid popularity bias amplification, we separate items into ten groups based on their popularity and aim to align the popularity distribution of the recommended items with that of the training set.

*4.1.2 Baselines.* We evaluated MoRec against several competitive scalarization and Pareto multi-objective learning methods:

- **Static**: The static method combine different objectives by a fixed weight, with different solutions generated by assigning varying weights. Recent research has shown that the true potential of static linear scalarization has been underestimated by literature [45].
- **MGDA** [10]: It aims to find a common descent direction for all the objectives by solving a convex quadratic programming problem that minimizes the norm of the weighted sum of the gradients of each objective. To generate various solutions with MGDA, one could modify the random seed.
- **PEMTL** [23]: PEMTL is an extension of MGDA, which can generated distributed Pareto-efficient solutions by adding extra constraints to the quadratic programming problem. Various solutions could be generated by setting different preference vectors.
- **EPO** [26]: EPO further enhances PEMTL by proposing a novel gradient combination method that aims to find an extract solution consistent with objective preference vector.

Typically, these methods require the objective function to be differentiable. The revenue objective is easily differentiable, as the loss function is weighted by the profit of the clicked item, as shown in Eq.(15). However, constructing a data sample-wise differentiable objective for the alignment and fairness objective requires some tricks. For alignment, the loss function is weighted by the reciprocal of the clicked item's popularity, as shown in Eq.(16). For fairness, we use the Pearson correlation as a regularization term, similar to [3], as shown in Eq.(17). Consequently, the baselines can optimize all four objectives. Formally,

$$L_{rev} = \sum_{(u,i)} r(i) \cdot \ell(u, i) \qquad (15)$$

$$L_{ali} = \sum_{(u,i)} \frac{1}{\text{pop}(i)} \cdot \ell(u, i) \qquad (16)$$

$$L_{fai} = \frac{\left( \sum_i \hat{y}_\theta(x_i) - \mu_{\hat{y}_\theta} \right) \left( \sum_i g_i - \mu_g \right)}{\sigma_{\hat{y}_\theta} \sigma_g} \qquad (17)$$

where $\hat{y}_\theta$ and $g_i$ denote the prediction of model and the sensitive attribute of sample $x_i$ (i.e. the item category), $\mu_*$ and $\sigma_*$ represents the mean and standard deviation operation, respectively.

**Table 1: Dataset Statistics.**

|  | #users | #items | #interactions |
|---|---|---|---|
| Electronics | 124,917 | 44,848 | 1,072,840 |
| Movies | 89,922 | 38,563 | 1,146,563 |
| Xbox | 154,210 | 5,161 | 6,058,454 |

*4.1.3 Evaluation Metrics.* For accuracy, we employ the widely adopted Hit metrics. To assess fairness, we utilize the definition in Eq.(6) and adopt the least misery metric [43] (in terms of Hit measure), as our evaluation standard. To evaluate alignment, we use KL-divergence denoted in Eq.(9) as measure, by setting $P(x_i, \theta), Q(x_i)$ to the frequency distribution of recommended items and a specified popularity distribution, respectively. A smaller Pop-KL value indicates better alignment performance in the model's recommendations. For revenue assessment, we rely on price-weighted Hit as the primary evaluation criteria, termed rHit, as defined in [22]. Higher values of these metrics correspond to a greater revenue potential for the model's predictions. All metrics are calculated based on the top-10 recommendations. Additionally, we calculate the average relative improvements comparing to the base model across all objectives, serving as a criteria for solution selection, abbreviated as *Imp*.

*4.1.4 Implementation Details.* To verify our framework is model-agnostic, we use two different base models for experiments: MF-BPR [30] and SASRec [20]. The embedding dimension of both base models is set to 64. Other model parameters, such as the number of transformer layers and the number of attention heads, remain consistent with the original paper. Regarding the training procedure, we utilize Adam as the optimizer and set the learning rate to 0.001. The batch size and weight decay are tuned within the sets 512, 1024 and 0, $10^{-6}, 10^{-5}$, respectively. For SASRec, we use the binary cross-entropy loss to keep consistent with the original paper. The number of negative samples is set to 10 and 3 for MF-BPR and SASRec, with negatives sampled according to the distribution of item popularity proposed in [29]. For the objective coordinator, the $\alpha, \lambda, K_p, K_i$ values are empirically set to 0.1, 0.2, 0.01 and 0.001, respectively. The expected loss value varies across datasets and backbone models. For MF-BPR, the expected loss values $\hat{\ell}$ are set to 0.20, 0.20, and 0.55 for Electronics, Movies, and Xbox, respectively. For SASRec, the expected loss values are set to 0.22, 0.22, and 0.24 for Electronics, Movies, and Xbox, respectively. MGDA[3], PEMTL[4] and EPO [5] are implemented with the source code. All experiments are conducted on a single Nvidia A100 based on Pytorch 1.12 framework.

We pretrain the backbone model until convergence. Then, we apply all multi-objective methods to the well-trained model for continual training, maintaining the same training parameters. The continual training process concludes when the model converges.

## 4.2 Overall Performance

We first examine how effective is our proposed model for simultaneously optimizing four objectives. We assume that an effective

---

[3]https://github.com/isl-org/MultiObjectiveOptimization
[4]https://github.com/Xi-L/ParetoMTL
[5]https://github.com/dbmptr/EPOSearch

method should jointly optimize multiple objectives without significantly compromising the accuracy performance. Consequently, we deem a solution to be ***invalid*** if it exhibits less than 97% accuracy compared to the base model. We generate at least six solutions for each baseline as well as our model. The most optimal solution is selected from all valid solutions based on the average relative improvement *Imp*. If all solutions are invalid, we opt for the one with the highest accuracy. Results are presented in Table 2 and Table 3.

All the baseline methods exhibit a relative improvement in certain objectives compared to the base model. When the base model is MF-BPR, all the baseline methods display significant average relative enhancements, despite some invalid solutions, underscoring the effectiveness of the scalarization method when the base model is not robust. However, for the more complex SASRec model, it becomes challenging to demonstrate strong overall enhancements, and some methods may even result in negative overall improvements.

On the other hand, MoRec can enhance the performance on target objectives with minimal accuracy loss, particularly on the industrial dataset. Specifically, MoRec achieves a maximum relative accuracy drop of 2.76% over three datasets, highlighting the effectiveness of our PID-based coordinator. Notably, none of the baseline models can be controlled to ensure a valid solution in different settings. Additionally, MoRec outperforms all the baseline methods in terms of average enhancements. While MoRec does not achieve state-of-the-art (SOTA) results for some individual objectives, its performance remains competitive, as it is close to the SOTA individual. Moreover, MoRec is the only model among its competitors that effectively performs on both types of base models, highlighting its superiority in terms of model-agnostic properties.

## 4.3 Pareto Efficiency Study

To validate the Pareto efficiency of MoRec, we generate five solutions for each method and draw the Pareto Frontier. For better visualization, we follow the two-objective setting by optimizing accuracy and revenue/fairness on two public datasets. The results are shown in Figure 3.

We observe that our MoRec exhibits great Pareto efficiency in all the four cases, especially in Electronics. While baseline methods fail to demonstrate Pareto efficiency in fairness, the reason may lie in that the loss for fairness is not designed for optimizing least misery directly and heterogeneous with accuracy loss. Furthermore, the solutions generated by MoRec have lower drop rates in accuracy and even obtain slight improvements, suggesting that PID-based objective coordinator's capability in controlling the degradation of accuracy. As for baselines, we indeed have the similar observation with [22] that solutions generated by MGDA are more centralized compared with PEMTL and EPO.

## 4.4 Ablation Study

To investigate the importance of the proposed adaptive data sampler (DS) and PID-based objective coordinator (OC), we conduct ablation studies under the two-objective setting on the Electronics dataset, with the results presented in Figure 4. In MoRec w/o DS, we replace the data sampler with the extra loss function in Eq.(15) to model

**Table 2: Performance over four objectives with MF-BPR. Bold and underline represent the best and second best results, respectively. ↓ represents the performance of accuracy drops more than 3% compared with Base. Numbers are in percentage.**

| Dataset | Electronics | | | | | Movies | | | | | Xbox | | | | |
|---|---|---|---|---|---|---|---|---|---|---|---|---|---|---|---|
| Metrics | Hit | rHit | Pop-KL | min-Hit | *Imp* | Hit | rHit | Pop-KL | min-Hit | *Imp* | Hit | rHit | Pop-KL | min-Hit | *Imp* |
| Base | 1.62 | 135.42 | 142.54 | 0.91 | 0.00 | 4.09 | 112.68 | 83.11 | 3.57 | 0.00 | 20.27 | 532.53 | 51.94 | 3.28 | 0.00 |
| Static | 1.62 | 197.56 | 37.09 | 1.00 | 32.41 | 4.06 | 136.92 | 27.74 | 2.16 | 11.99 | 18.22↓ | **681.73** | 4.93 | 3.68 | *Invalid* |
| MGDA | 1.32↓ | **262.84** | 20.37 | 0.32 | *Invalid* | 3.90↓ | **179.71** | 9.32 | 3.23 | *Invalid* | 14.38↓ | 418.16 | 11.33 | 9.05 | *Invalid* |
| PEMTL | 1.68 | 167.53 | 99.08 | 1.03 | 17.60 | 4.08 | 161.25 | **9.09** | 3.10 | 29.70 | 17.54↓ | 679.04 | 6.34 | 3.37 | *Invalid* |
| EPO | 1.51↓ | 162.99 | 35.75 | 0.98 | *Invalid* | 3.97 | 160.46 | 9.14 | 2.44 | 24.22 | 16.89↓ | 645.64 | **3.89** | 4.90 | *Invalid* |
| MoRec | 1.63 | 225.19 | **16.81** | **1.05** | **42.60** | 3.98 | 164.44 | 9.73 | **3.68** | **33.69** | 19.71 | 575.66 | 18.52 | **11.98** | **83.79** |

**Table 3: Performance over four objectives with SASRec-BCE. Bold and underline represent the best and second best results, respectively. ↓ represents the performance of accuracy drops more than 3% compared with Base. Numbers are in percentage.**

| Dataset | Electronics | | | | | Movies | | | | | Xbox | | | | |
|---|---|---|---|---|---|---|---|---|---|---|---|---|---|---|---|
| Metrics | Hit | rHit | Pop-KL | min-Hit | *Imp* | Hit | rHit | Pop-KL | min-Hit | *Imp* | Hit | rHit | Pop-KL | min-Hit | *Imp* |
| Base | 1.81 | 174.53 | 26.38 | 0.75 | 0.00 | 5.93 | 175.17 | 10.78 | 4.13 | 0.00 | 25.99 | 809.74 | 17.16 | 7.84 | 0.00 |
| Static | 1.84 | **259.75** | 20.49 | 0.40 | 6.51 | 5.21↓ | 163.30 | 8.63 | 3.24 | *Invalid* | 13.73↓ | 700.68 | 32.45 | 0.67 | *Invalid* |
| MGDA | 1.70↓ | 175.00 | 42.23 | 0.37 | *Invalid* | 5.50↓ | 167.22 | 14.34 | 4.57 | *Invalid* | 25.66 | 932.78 | 20.88 | 8.46 | 4.00 |
| PEMTL | 2.46 | 220.91 | 45.40 | 1.28 | 15.10 | 5.81 | 153.64 | 14.69 | 4.08 | -12.94 | 25.41 | 812.74 | 32.84 | 9.63 | -17.58 |
| EPO | 1.69↓ | 228.42 | 43.57 | 0.03 | *Invalid* | 5.15↓ | 157.85 | 18.05 | 3.24 | *Invalid* | 21.12↓ | **995.70** | 47.42 | 2.03 | *Invalid* |
| MoRec | 2.32 | 239.54 | **14.87** | **1.47** | **51.38** | 6.25 | **189.26** | **1.64** | **5.17** | 30.84 | 25.96 | 899.47 | **7.64** | **14.12** | **36.65** |

revenue. As for the objective coordinator, we replace our PID-based OC with various baseline methods, denoted by the pattern *MoRec-OC-xx*. Similar with the setting in Section 4.3, we generate five solutions for each variant for visualization.

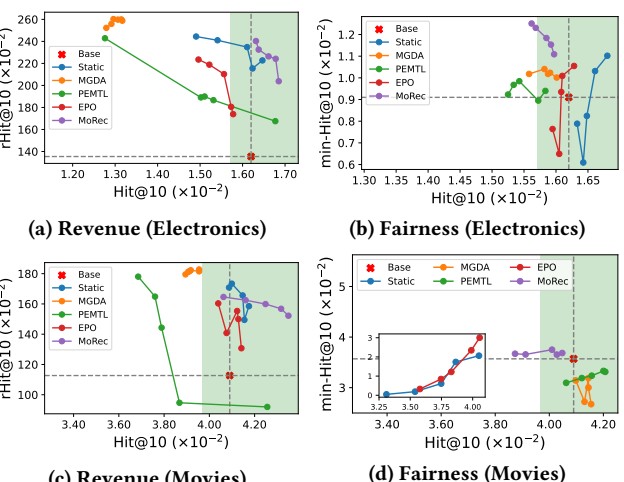

**(a) Revenue (Electronics)**   **(b) Fairness (Electronics)**

**(c) Revenue (Movies)**   **(d) Fairness (Movies)**

**Figure 3: Study of Pareto efficiency over two objectives. The green part represents that the accuracy drop is less than 3% compared to base.**

First, when DS is replaced (MoRec w/o DS), the frontier in the revenue scenario remains competitive Pareto efficiency, which is not surpassed by MoRec. The reason lies in that our DS draws samples in proportion to their revenue, which is approximately equal to the weighted loss in $L^{rev}$. Nonetheless, due to the heterogeneity between accuracy and fairness loss functions, MoRec w/o DS

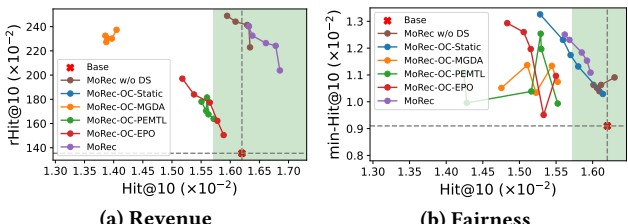

**(a) Revenue**   **(b) Fairness**

**Figure 4: Ablation Study. The green part represents that the accuracy drop is less than 3% compared to base.**

exhibits weaker Pareto efficiency in the fairness scenario. In both scenarios, the accuracy performances are guaranteed due to the PID controller. Second, the replacement of our OC results in a failure to control accuracy performance, as most of the solutions from the variants fall outside the green area, as observed. Moreover, all the variants exhibit weaker Pareto efficiency compared to MoRec, especially in Figure 4b, which underscores the indispensability of the OC component.

## 4.5 Control Effect

With a unified objective modeling and a PID-based objective coordinator, MoRec demonstrates a strong control effect on objective preference. To verify this, we visualize the PID's controlling ability under the two-objective setting in Section 4.3. We plot the accuracy loss, Hit, and rHit curves during the training stage in Figure 5. We observe that the PID-based coordinator can effectively regulate the loss value to an expected value. The metrics on accuracy and loss exhibit an inverse relationship, meaning that the higher the expected loss value, the lower the corresponding accuracy - which

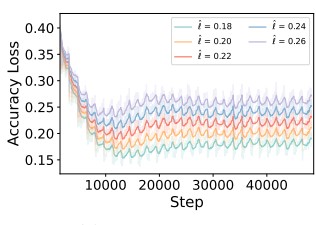

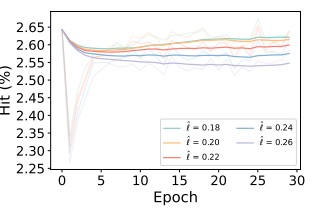

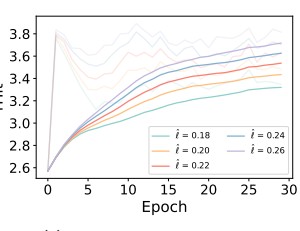

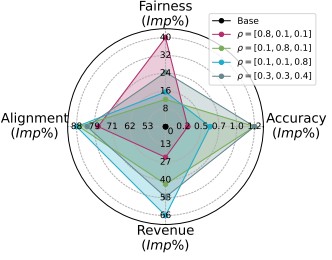

(a) Accuracy Loss    (b) Accuracy: Hit@10    (c) Revenue: rHit@10

Figure 5: Visualization of the precise control effect of our PI Controller in validation set.

Figure 6: Visualization of objective preference control effect.

is expected. In contrast, the metric on revenue increases as the expected value of accuracy loss rises, which is also expected.

Finally, we verify whether MoRec can flexibly control solutions' generation towards specific objective preferences in four-objective setting. By setting various objective preference vectors $\rho = [\rho_{fai}, \rho_{ali}, \rho_{rev}]$, we can obtain diverse solutions. Results are illustrated in Figure 6. Numbers in the figure represent the relative improvement compared to the base model. We observe a strong pattern in the relation between $\rho$ and the resulting objectives. For instance, the preference coefficient $\rho = [0.8, 0.1, 0.1]$ prioritizes the fairness metric, so its solution has a higher min-Hit than the others; preference coefficient $\rho = [0.3, 0.3, 0.4]$ leads to a relatively more balanced solution among objectives.

## 5 RELATED WORK

### 5.1 Accuracy-oriented Recommendation

Classical recommendation algorithms primarily focus on improving prediction accuracy by optimizing accuracy-oriented loss functions, such as Mean Square Error (MSE), Bayesian Personalized Ranking loss (BPR[30]), Binary Cross Entropy loss (BCE), and log-softmax loss. Depending on data types and patterns in various application scenarios, numerous backbone models have been proposed to enhance the accuracy of recommender systems. For instance, matrix factorization (MF)[16, 19, 27, 30] mainly focuses on latent user-item interactions for collaborative filtering learning; deep neural network-based approaches [6, 14, 15, 21, 37, 39] are employed for deep feature interactions; and sequential recommendation techniques[17, 20, 38] capture the order of user behavior history. In contrast to this line of research, the goal of this paper is not to propose a new backbone model, but rather to introduce a novel learning framework, enabling a given backbone model to be optimized for multiple diverse objectives simultaneously.

### 5.2 Multi-Objective Problem

Multi-objective optimization (MOP) aims at finding a set of Pareto solutions with different trade-offs, with origins dating back to the early 1900s[11]. Multi-objective optimization methods can be broadly classified into two categories: multi-objective evolutionary algorithms (MOEAs) and scalarization. MOEAs[9, 12, 34] employ various population-based heuristic search techniques to obtain solutions that are not dominated by each other, albeit at a high time cost. Scalarization methods [5, 47] transform MOPs into single-objective problems (SOPs), with the weighted sum being the most

commonly used technique. In order to obtain the Pareto efficiency, the multiple-gradient descent algorithm [10] combines scalarization with stochastic gradient descent (SGD), using the KKT condition to update the weights. MGDA[35] is later improved to solve multi-task problems using the Frank-Wolfe algorithm. However, MGDA does not have a systematic way to incorporate various priorities. Recent works PEMTL[23] and EPO[25] present methods for generating solutions tailored to specific preferences by adding extra constraints to the quadratic programming problem.

Multi-objective recommendation (MOR) aims to optimize multiple objectives simultaneously within a joint recommendation framework[18, 48]. MOEAs are designed with different heuristic search[4, 7, 28, 31, 32, 40, 50] or model hybridization[4, 31, 32] strategies in balancing accuracy, diversity, long-tail performance, et al, which usually regard recommendation lists or well-trained models as solutions and do variation to generate new solutions. Especially, several scalarization methods are proposed in recent works. A two-step method[22] built upon MGDA optimizes CTR and GMV by relaxing the quadratic programming problem as a non-negative least squares problem. And a reinforcement learning-based strategy[44] is proposed to solve the minimization optimization problem in MGDA, aiming to balance CTR and dwell time. However, existing methods only consider two or three homogeneous objectives without priorities and focus on modeling different objectives with various well-designed loss functions.

## 6 CONCLUSION

In this paper, we emphasizes the significance of multi-objective recommendation and consolidate various objectives into four fundamental forms, paving the way for a more systematic and coherent understanding of multi-objective optimization in recommender systems. We introduce a novel and model-agnostic MoRec framework for multi-objective recommendation, which features a tri-level structure comprising an adaptive data sampler and a PID-based objective controller. Our MoRec framework presents a flexible and adaptable solution for real-world applications, allowing for improved performance across multiple objectives without requiring modifications to existing model architectures or optimizers. Through extensive experiments conducted on three real-world datasets, we demonstrate the effectiveness and superiority of the MoRec framework. The results of this study contribute to the development of more efficient and adaptable recommender systems, fostering further exploration and advancement in multi-objective optimization techniques.

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

Received 20 February 2007; revised 12 March 2009; accepted 5 June 2009

