# OpenReview forum: "A Data-Centric Multi-Objective Learning Framework for Responsible Recommendation Systems"
_ACM.org/TheWebConf/2024/Conference — TheWebConf24_

### Official Review · Reviewer_rBbM · 2023-11-19

**Novelty:** 5
**Technical Quality:** 5

**Review:**

Summary:

The authors propose to reweight the training samples for different recommendation objectives (specifically, accuracy, revenue, fairness, and alignment), and manually reweight the training loss for different objectives for controllable multi-objective learning. They also employ PID to ensure the performance of the key objective. The method is named MoRec. The authors verify the method on three public datasets.

Strength:

1. Multi-Objective Recommendation is a very valuable research topic.
2. The proposed method is technically sound and it is easy to be applied in real industrial applications.

Weakness:

1. The baselines and related works in the paper appear to be somewhat inadequate. The results of some basic/advanced multi-task learning methods are curious.

E.g. (1) Xie R, Liu Y, Zhang S, et al. Personalized approximate Pareto-efficient recommendation[C]//Proceedings of the Web Conference 2021. 2021: 3839-3849.

(2) Chen S, Wang Y, Wen Z, et al. Controllable Multi-Objective Re-ranking with Policy Hypernetworks[C]//Proceedings of the 29th ACM SIGKDD Conference on Knowledge Discovery and Data Mining. 2023: 3855-3864.

(3) Yang E, Pan J, Wang X, et al. Adatask: A task-aware adaptive learning rate approach to multi-task learning[C]//Proceedings of the AAAI Conference on Artificial Intelligence. 2023, 37(9): 10745-10753.

(4) Kendall A, Gal Y, Cipolla R. Multi-task learning using uncertainty to weigh losses for scene geometry and semantics[C]//Proceedings of the IEEE conference on computer vision and pattern recognition. 2018: 7482-7491.

2. Online systems can also achieve trade-offs and optimization of different goals through personalized distribution of traffic. The paper should explain the relationship between its work and these works.

E.g. Liu X, Yu C, Zhang Z, et al. Neural auction: End-to-end learning of auction mechanisms for e-commerce advertising[C]//Proceedings of the 27th ACM SIGKDD Conference on Knowledge Discovery & Data Mining. 2021: 3354-3364.

**Questions:**

1. How to understand "Responsible" recommendation systems?
2. Industrial recommendation systems may have several specific objectives such as "watching time", "positive action number", etc. Thus, the proposed objectives in the paper, namely accuracy, revenue, fairness, and alignment may not be very suitable for all the recommendation systems. The paper gave a practice of MoRec for optimizing these four objectives. So I wonder if MoRec can address the optimization of other metrics. If so, can you give a formulation of the metric that can be handled by MoRec, (such as differentiable metrics, just an example) and a unified one? If the MoRec is ad-hoc for these metrics, the generalization is a little limited.
3. Alignment is an interesting topic in recommendation and advertising systems. The paper adopts KL-divergence to align the model's distribution with some pre-defined expectation distribution. I think it's suitable for recommendation systems but does not fit well with advertising systems that wish to align the predictions with true probability values. Thus adopting the metrics of uncertainty calibration such as $ECE$,  $ECE_{sweep}$(ref. Roelofs R, Cain N, Shlens J, et al. Mitigating bias in calibration error estimation[C]//International Conference on Artificial Intelligence and Statistics. PMLR, 2022: 4036-4054.), $MVCE$(ref. Huang S, Wang Y, Mou L, et al. MBCT: Tree-Based Feature-Aware Binning for Individual Uncertainty Calibration[C]//Proceedings of the ACM Web Conference 2022. 2022: 2236-2246.) would extend the MoRec framework to a more general version. I wonder that can MoRec adapt to these metrics easily. Because the paper is focused on recommendation systems, I make this question independent from question 2. The results do not affect the self-consistency of the paper.

**Reviewer Confidence:**

3: The reviewer is confident but not certain that the evaluation is correct

**Scope:**

4: The work is relevant to the Web and to the track, and is of broad interest to the community

---

### Official Review · Reviewer_QR5b · 2023-11-24

**Novelty:** 2
**Technical Quality:** 3

**Review:**

The paper introduces a tri-level framework designed to unify different objective functions in recommendation systems, with experimental validation of its effectiveness. However, several concerns are raised:
1. The baselines used for comparison are relatively old, which undermines the ability to clearly demonstrate the improvements and contributions of the proposed framework.
2. The two base models employed are not the latest state-of-the-art (SOTA) methods, casting doubt on the framework's relevance and utility in the current landscape of recommendation systems.
3. Many of the objectives seem to be directly applied from existing works, which diminishes the perceived originality of the research. The motivation behind the authors' alignment objective, aimed at addressing skew distribution, appears insufficiently substantiated. Furthermore, there is a lack of detailed analysis in the experiments on how the alignment objective effectively addresses this issue.
4. The paper lacks an in-depth analysis of complexity and theoretical guarantees. Reliance solely on experimental comparisons makes it challenging to convincingly demonstrate the practicality of the tri-level framework.
5. The work is said to be inspired by FairBatch, yet there is no detailed performance comparison between MoRec and FairBatch within the manuscript. This omission raises questions about the distinctiveness and advancement over FairBatch.
6. The paper mentions its relevance to Multiple Objective Optimization but fails to compare with significant related works, such as references [22, 24]. This lack of comparison with important literature limits the contextual understanding of the framework's contribution.
7. The paper does not include the most recent references from 2023, which is concerning for a field that rapidly evolves. This raises questions about the currency and relevance of the literature review.

**Questions:**

Mentioned in Review.

**Reviewer Confidence:**

3: The reviewer is confident but not certain that the evaluation is correct

**Scope:**

4: The work is relevant to the Web and to the track, and is of broad interest to the community

---

### Official Review · Reviewer_5DeY · 2023-11-24

**Novelty:** 4
**Technical Quality:** 4

**Review:**

Authors in this paper proposed MoRec which is a tri-level and unified framework for multi-objective recommendations.

Pros
- the topic is relevant
- re-weighting strategy is used to help achieve a unified framework


Cons
- it's interesting to see such a framework or library proposed for MoRec. The framework considers accuracy, revenue, fairness and alignment. However, it may not be flexible, since not all the data allows us to compute the revenue, such as researchers in academica may not have access to revenue data. Authors may figure out an alternative to represent "revenue" in a general/common data, such as MovieLens, even if actual revenue data is not available.
- Authors used scalarization method to setup the objectives, where the objective weights are considered as hyper-parameters. In MoRec, decision makers' preferences on objectives are usually unknown, which results in difficulties in seting up these weights and helping them to find a single optimal solution. By setting them as hyper-parameters, users have to try multiple attempts, and they may skip some solutions in Pareto front. Authors may need to figure out some ways to help users search for or find these weights more efficiently. Also, this challenge is directly related to how to find the optimal solution (i.e., a single solution) from Pareto set.
- I think this paper is more like a submission in demo/library session, rather than a significant contribution in full paper session.

**Questions:**

how to help users find the optimal solution? without known weights on objectives from decision makers?

**Reviewer Confidence:**

4: The reviewer is certain that the evaluation is correct and very familiar with the relevant literature

**Scope:**

3: The work is somewhat relevant to the Web and to the track, and is of narrow interest to a sub-community

---

### Official Review · Reviewer_xiUW · 2023-11-26

**Novelty:** 5
**Technical Quality:** 6

**Review:**

The paper is well written, clear and addresses a common need in the recommendation space in terms of multi-objective learning. It builds on the FairBatch approach, effectively extending it from two considerations to four.  The approach is data centric, making it model agnostic, which makes the work complementary to a number of approaches and relatively easy to test in a commercial setting.  As, such the general approach can be good for a framework.

The authors outlines the use of a PI controller in tri-level optimization framework, “MoRec”.  The outline of the methodology is clear, as is the experimental design. The experimental selection is interesting in that more than one type of scenario is chosen (i.e. movies, where there is little cost with a selection, and purchases, where there is).  The results are well presented and show the value of the work and “MoRec”.   The authors demonstrate value of their approach and the results are overall interesting.

As such, the paper is interesting, but it falls short of being a framework.

The authors did not consider other methodologies or approaches. The review of other approaches is not exhaustive, and does not mention how their methodology compares to other successful approaches, such as a metalearner.  In the references on a multi-objective recommendation system, I would have expected references to Burke, Abdollahpouir and perhaps Nguyen.

There were significant assumptions made in some cases without explaining the rationale.  For example, the authors claim “inherent conflicts among different objectives”, but this is not always the case.  Some objectives can be significantly aligned, others not, and that changes how to frame the problem and solve the objective functions.  This is not addressed.

Also, in the reference to the four fundamental forms, there is the potential for more. The list is not complete and hence not foundational. For example, it is missing short term versus long term customer satisfaction, and risk.  This is a significant concern for recommendations where there is significant cost and the timing between selection and consumption is significant.  For example, travel, when you take a significant risk on a trip by booking a hotel for an extended period you do not have direct experience with and little recourse once you arrive.  As such, it would have been good for the authors to outline a method to add to the objectives, rather as present four fixed ones that may not be relevant to multiple commercial applications.

The revenue estimation foundation is estimated only using an item’s price, which is a very coarse selection. To the degree the authors want to offer a framework, this is very limiting. Margin levels can vary significantly between products and are core to recommendations.  The classic McDonald’s meal plans are based on a play on margins (i.e. include a high margin drink and fries with low margin meat sandwich).  How would subscription services be considered too?

It mentions, “current methods (for multi-objective learning) two challenges” but does not go into their detail other than listing them.  It is not clear that there are other methods to overcome those challenges other than what the authors propose.

To the degree the revenue approximation can be expanded and the four objectives made expandable/exchangeable the methodology will have more applicability and can be considered a framework.

**Questions:**

It would be good to address the points in the review.

Specifically,
- How would you compare your methodology and results with those of a meta learner (or other approach for multi-objective)?
- How would you add more criteria, both conflicting and complementary
- Can you expand to include a more flexible (and more applicable) treatment of revenue?

**Ethics Review Description:**

No issues

**Reviewer Confidence:**

3: The reviewer is confident but not certain that the evaluation is correct

**Scope:**

3: The work is somewhat relevant to the Web and to the track, and is of narrow interest to a sub-community

---

### Decision · Program_Chairs · 2024-01-22

**Decision:**

Accept

**Comment:**

With the exception of one reviewer, who mainly focuses on novelty concerns, the reviewers were generally positive about this work, both in terms of novelty and technical quality. A number of concerns were raised, though, e.g., regarding alternative baselines and other existing methods, to which the authors provided detailed responses. Questions were also raised if the technical contribution is sufficient for a research paper track. Unfortunately, the authors only promise to share the code and data after acceptance. This makes it impossible for reviewers to assess the reproducibility level of this research in the review phase, and it is a pity as the paper actually proposes a framework that is intended for others to use.